# Origami: Single-cell 3D shape dynamics oriented along the apico-basal axis of folding epithelia from fluorescence microscopy data

Tania Mendonca[1,2¤]*, Ana A. Jones[2], Jose M. Pozo[1,3], Sarah Baxendale[2], Tanya T. Whitfield[2], Alejandro F. Frangi[1,3,4]*

1 Centre for Computational Imaging and Simulation Technologies in Biomedicine (CISTIB), Department of Electronic and Electrical Engineering, University of Sheffield, Sheffield, United Kingdom, 2 Department of Biomedical Science, Bateson Centre and Neuroscience Institute, University of Sheffield, Sheffield, United Kingdom, 3 Centre for Computational Imaging and Simulation Technologies in Biomedicine (CISTIB), School of Computing and School of Medicine, University of Leeds, Leeds, United Kingdom, 4 Medical Imaging Research Center (MIRC), University Hospital Gasthuisberg, Cardiovascular Sciences and Electrical Engineering Departments, KU Leuven, Belgium

¤ Current address: Optics and Photonics Research Group, Department of Electrical and Electronic Engineering, University of Nottingham, Nottingham, United Kingdom
* Tania.Mendonca@nottingham.ac.uk (TM); A.Frangi@leeds.ac.uk (AFF)

**Data Availability Statement:** Code can be found on GitHub: https://github.com/cistib/origami Data

## Abstract

A common feature of morphogenesis is the formation of three-dimensional structures from the folding of two-dimensional epithelial sheets, aided by cell shape changes at the cellular-level. Changes in cell shape must be studied in the context of cell-polarised biomechanical processes within the epithelial sheet. In epithelia with highly curved surfaces, finding single-cell alignment along a biological axis can be difficult to automate *in silico*. We present 'Origami', a MATLAB-based image analysis pipeline to compute direction-variant cell shape features along the epithelial apico-basal axis. Our automated method accurately computed direction vectors denoting the apico-basal axis in regions with opposing curvature in synthetic epithelia and fluorescence images of zebrafish embryos. As proof of concept, we identified different cell shape signatures in the developing zebrafish inner ear, where the epithelium deforms in opposite orientations to form different structures. Origami is designed to be user-friendly and is generally applicable to fluorescence images of curved epithelia.

## Author summary

During embryonic development, two-dimensional epithelial sheets bend and fold into complex three-dimensional structures–like paper in the origami art form. The genetic and biomechanical processes driving epithelial folding can be polarised in the epithelium, leading to asymmetric shape changes at the single cell level. Defects in such epithelial shaping have been linked to many developmental anomalies and diseases. It is, therefore, important not only to quantify shape change at the single cell level, but also to orientate these asymmetrical changes along an epithelial axis of polarity when studying morphogenetic processes. Origami is a MATLAB-based software that has been developed to

can be found on FigShare: https://doi.org/10.6084/m9.figshare.14531421.v1.

**Funding:** Work was supported by a Biotechnology and Biological Sciences Research Council (BBSRC; https://bbsrc.ukri.org/) project grant to TTW, SB and AFF (BB/M01021X/1). Imaging was carried out in the Wolfson Light Microscopy Facility at the University of Sheffield, supported by a Biotechnology and Biological Sciences Research Council BBSRC ALERT14 award to TTW and SB for light-sheet microscopy (BB/M012522/1). AAJ was funded by a Doctoral Training Award from the Biotechnology and Biological Sciences Research Council BBSRC White Rose Doctoral Training Partnership in Mechanistic Biology (BB/M011151/1). AFF is supported by the Royal Academy of Engineering Chair in Emerging Technologies Scheme (CiET1819\19) and the MedIAN Network (EP/N026993/1) funded by the Engineering and Physical Sciences Research Council (EPSRC; https://epsrc.ukri.org/). The funders had no role in study design, data collection and analysis, decision to publish, or preparation of the manuscript.

**Competing interests:** The authors have declared that no competing interests exist.

automatically extract such single-cell asymmetrical shape features along the epithelial apico-basal axis from fluorescence microscopy images of folding epithelia. Origami provides a solution to computing directional vectors along the epithelial apico-basal axis followed by extracting direction-variant shape features of each segmented cell. It is generally applicable to epithelial structures regardless of complexity or direction of folding and is robust to imaging conditions. As proof of concept, Origami successfully differentiated between different cell shape signatures in highly curved structures at different developmental timepoints in the zebrafish inner ear.

This is a *PLOS Computational Biology* Software paper.

## Introduction

Complex morphologies across taxa and tissue types are generated through the deformation of epithelial sheets [1–3]. In the embryo, many developing epithelia form highly curved surfaces. Epithelial folding processes are driven by polarised mechanical forces and involve three-dimensional changes in shape at the cellular level [4,5]. Fluorescence imaging techniques have made it possible to follow such shape changes at cellular resolution, *in vivo* and in real-time [6–8]. These imaging advances have consequently driven the development of tools to quantify epithelial dynamics, especially cell shape changes.

Many image analysis tools measuring cell shape change have been limited to two-dimensional [9–12] or quasi-3D fluorescence microscopy data [13]. Extending these measurements to 3D has been aided by the development of membrane-based 3D segmentation methods such as ACME [14], RACE [15], 3DMMS [16], CellProfiler 3.0 [17], and more recently, deep-learning-based methods [18–21]. Some image analysis tools, such as CellProfiler 3.0 [17], Morpho-GraphX [22] and ShapeMetrics [23], provide pipelines to compute direction-invariant cell shape features. However, finding the position of 3D-segmented cells along biologically relevant axes to quantify directional shape features is still a challenging problem that has so far not seen a generalised solution.

Solving the orientation of individual cells relative to the known overall polarity of the epithelial sheet is critical, as cell-polarised biomechanical processes drive cell shape changes; constriction or expansion can occur along either the apical [24,25] or baso-lateral [26] cell surfaces and can be detected by any skew in mass distribution within a cell along an apico-basal axis of symmetry. Epithelial folding may be initiated or influenced by cell proliferation, cell death, cytoskeletal remodelling, or changes in cell surface properties [27,28]. These mechanisms can lead to changes in cell shape features, including cell height and width, volume, surface area and sphericity.

Cell orientation or polarity can be defined along the plane of the epithelium (planar cell polarity) or perpendicular to the epithelial plane, along the apico-basal axis of the cell. Existing automated methods for assigning polarity to segmented cells often rely on additional biochemical markers for polarity [29–31]. Including such additional markers in fluorescence imaging experiments increases the time taken to generate each image, potentially leading to phototoxicity, and the resulting larger volume of image data makes analysis computationally expensive. Moreover, producing the required animals carrying multiple transgenes for live imaging can be challenging and costly. Some image analysis methods compute direction vectors for

individual cells by drawing normal vectors to polynomial functions, often ellipsoids, used to estimate the surface of the specimen—for example, entire embryos [15] or blastoderms [32] undergoing morphogenesis. These methods are specific to the geometry of the specimen and are unsuitable for analysing complex folded topologies at advanced morphologic developmental stages. A third method uses known features of cell shape to assign cell orientation, for example by applying principal component analysis (PCA) to compute the apico-basal axis in columnar cells in EDGE4D [33] and the anterior-posterior axis in zebrafish lateral line primordia using landmark-based geometric morphometrics [31], or orienting cells along their long axis in the zebrafish optic cup as in LongAxis [34]. These strategies will be applicable only if a dominant cell shape feature, for example cell height/width ratio, is known and remains unchanged over space and time.

We introduce a new automated and easy-to-use tool, Origami, for extracting direction-variant shape features along the apico-basal axis by reconstructing the epithelial surface using a triangular mesh (Fig 1). Origami applies to a wide range of geometries of specimens undergoing morphogenesis and automatically extracts direction vectors for individual cells aligned to the apico-basal axis of the epithelial sheet without requiring additional labels for polarity. Direction-variant shape features are calculated by computing the geometric moments for the volume enclosed by the polygon representation of each segmented cell [35]. We showcase the versatility of our method using data from an assortment of structures at a range of developmental stages within the otic vesicle (developing inner ear) of zebrafish embryos.

## Design and implementation

The Origami pipeline is preceded by a membrane-based segmentation step. For this, we employed the open-source ACME segmentation software [14]. The segmented data are subjected to two main operations within Origami; epithelial polarity direction vector assignment (Fig 1B) and extraction of shape features (Fig 1C).

## Assigning polarity direction to individual cells

To compute directionally variant cell shape features, such as skewness (asymmetry in cell mass), and longitudinal and transversal spread, the positioning of segmented cells must be found in 3D space along a biologically relevant axis—we chose the known apico-basal axis of the cell. The folding epithelium was reconstructed *in silico* as a thin 'crust'—an open surface mesh that triangulates the centroids of the segmented cells in 3D space using the Crust algorithm [36,37] (Fig 1B). The Crust method computes a surface mesh from unorganised points —cell centroids in our case, using the Voronoi diagram of the cell centroids.

Following this, our automated method corrects imperfections in the estimated surface mesh that can cause errors in the resulting direction vectors. The mesh is refined by removing duplications (in vertices or triangular faces computed) and any self-intersecting triangular faces. Non-manifold edges, that is, those edges shared by more than two triangular faces, are re-meshed as a manifold mesh using the ball-pivoting algorithm [38,39].

The triangular faces of the refined mesh are ordered, and so by applying the right-hand rule when generating normal vectors to the surface mesh, these vectors all point to the same side of the mesh representation of the epithelial surface (Fig 1B). At this point, there are still two possible opposing orientations for each computed vector—facing the apical or the basal face of the epithelium, with a difference only in sign. In the developing zebrafish otic vesicle, the apical surface of the epithelium faces the fluid-filled lumen of the vesicle [2,8,40]. We used this prior knowledge to inform the orientation of the vectors by setting the value of a binary orientation-determining parameter to 'in' so that they point to a convergent point which falls on the side

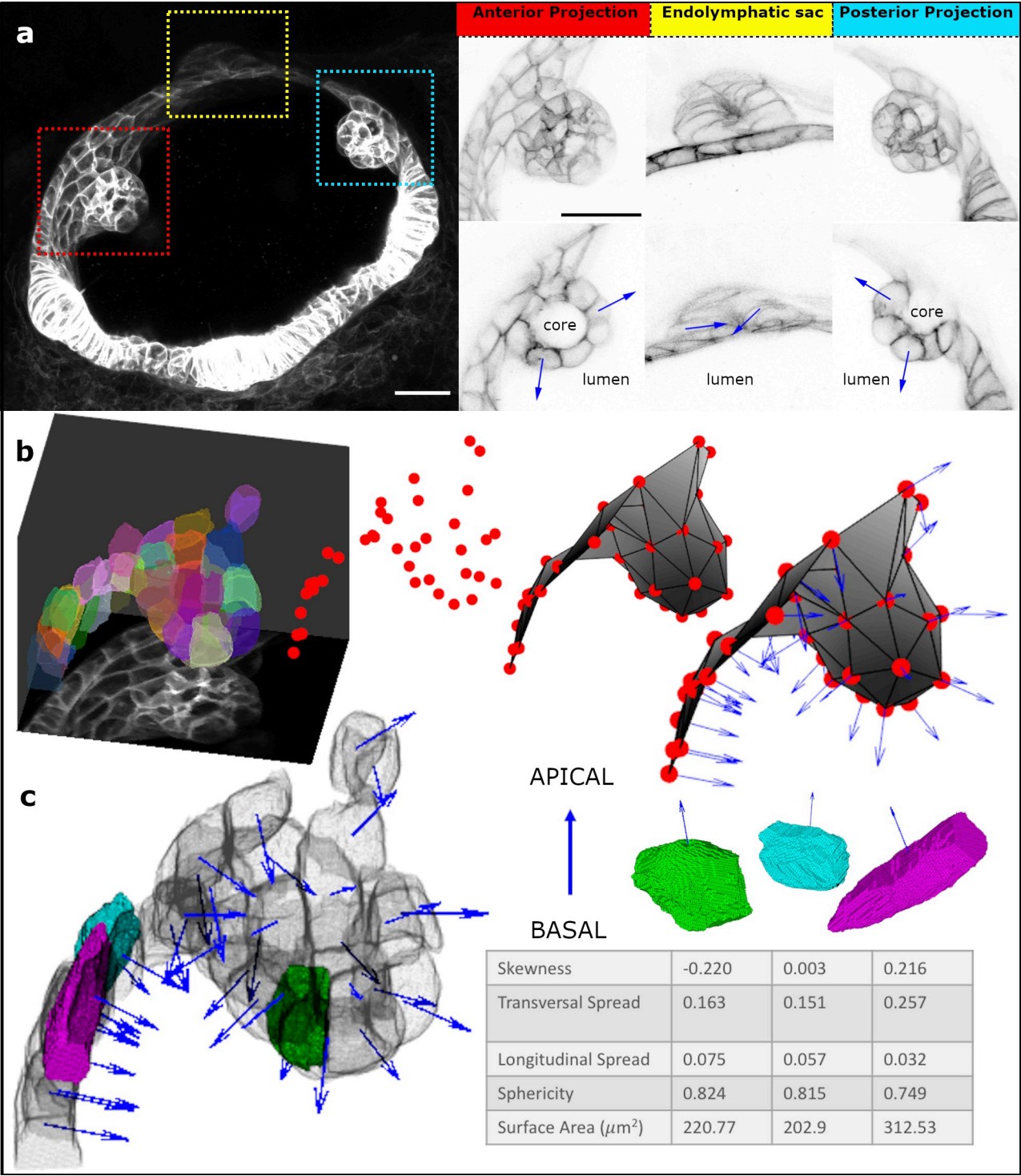

| | | | |
|---|---|---|---|
| Skewness | -0.220 | 0.003 | 0.216 |
| Transversal Spread | 0.163 | 0.151 | 0.257 |
| Longitudinal Spread | 0.075 | 0.057 | 0.032 |
| Sphericity | 0.824 | 0.815 | 0.749 |
| Surface Area ($\mu m^2$) | 220.77 | 202.9 | 312.53 |

**Fig 1. Origami Image Analysis Pipeline.** a. Airyscan confocal fluorescence micrograph (maximum intensity projection (MIP) of 35 z-slices) of the developing zebrafish otic vesicle at 51.5 hours post fertilisation. Red box—anterior projection; yellow box—endolymphatic sac; cyan box—posterior projection. The ROIs are expanded alongside—top row MIPs, and bottom row single slices. Scale bars: 20 μm. Blue arrows mark the direction of apicobasal polarity (pointing towards the apical side). b. Polarity assignment on segmented data; ROI surrounding the anterior projection was segmented (here overlaid on the MIP) using ACME, centroids were generated for each segmented cell and a triangular surface mesh was produced from these centroids. Normal vectors (blue arrows) to this surface mesh represent the apico-basal axis. c. Cell shape features were computed concerning the assigned apico-basal axis; here, three example cells are highlighted, alongside a 3D rendering showing their position in the anterior projection and the corresponding shape metrics in a table.

of the curved surface mesh that corresponds to the apical (lumenal) side of the epithelium at each cell. When a structure folds multiple times in opposing orientations, such as in the synthetic data generated for this study, 'in' sets the polarity direction vectors to point towards a global convergent point, in our case determined by the curvature of the whole synthetic epithelium.

Under-segmentation can cause missing regions or unwanted holes in the triangular mesh, introducing errors when ordering the triangular faces. Our pipeline attempts to repair these holes by detecting and then remeshing them where possible. Holes, when detected, are flagged as a warning to users about potential errors in the output. Normal vectors to the reconstructed surface represent the apico-basal axis of the epithelium and are generated for each segmented cell at their centroid position (Fig 1B and 1C).

## Computing shape features using 3D geometric moments

The shape of an object can be characterised using central geometric moments [41]. Geometric moments are widely used in object recognition and classification problems [42,43] since they (i) are simple to compute, (ii) organise features in orders of increasing detail, and (iii) can be extended to $n$ dimensions. Each moment, $G_{ijk}^{(V)}$, is defined by the integral over the object (in our case, each segmented cell), of the Cartesian coordinates monomial $x^i y^j z^k$, where $i, j, k \geq 0$, with the origin of coordinates at the centroid.

In our analysis pipeline, 3D geometric moments were computed using the algorithm introduced in [35]. The defining continuous integrals are exactly computed within the triangular surface mesh generated for each individual segmented cell, split into a sum:

$$G_{ijk}^{(V)} = \sum_{c \in Facets} sign(Vol_c) \int_{Tc} x^i y^j z^k \, dx \, dy \, dz,$$ (1)

where each tetrahedron $T_c$ is defined by a triangle in the surface mesh and the origin (cell centroid). The determinant gives the oriented volume of this tetrahedron,

$$Vol = \frac{1}{6} \begin{vmatrix} x_1 & x_2 & x_3 \\ y_1 & y_2 & y_3 \\ z_1 & z_2 & z_3 \end{vmatrix}.$$ (2)

Considering its sign, the determinant allows the algorithm to be applied to shapes of any complexity and topology. The integral in each $T_c$ is given by a closed formula involving only the Cartesian coordinates of the triangular vertices.

The geometric moments of low orders have simple, intuitive interpretations. The zero<sup>th</sup> order moment $G_{000}^{(V)}$ provides the volume of the object, here an individual cell. For central moments, the first order moments are trivially null: $G_{100}^{(V)} = G_{010}^{(V)} = G_{001}^{(V)} = 0$. The second-order moments correspond to the spread (covariance tensor) of the distribution. So, the projection of the mass of each cell along the corresponding polarity vector represents the 'spread' as variance in mass 'longitudinally' (along the apico-basal axis) and 'transversally' (along the epithelial plane). This allowed us to identify if cells were more or less columnar (tall cells) or squamous (flat cells) in shape. The third-order moments represent 'skewness', which is the deviation from symmetry. In our pipeline, skewness was measured along the polarity direction vector in the apico-basal direction, with positive skewness values indicating apical cell constriction and/or basal relaxation and negative values indicating basal cell constriction and/or apical expansion. A value of zero indicated no skew. Additionally, the sphericity of each cell

was computed as the ratio of the cell surface area to the surface area of a sphere with the same volume as the cell [44], from 0 for a highly irregularly-shaped cell to 1 for a perfect sphere.

## Results

### Evaluation of computed cell polarity direction vector

To evaluate the computed direction vectors denoting cell polarity, we generated 3D synthetic data representing curved, folding epithelia with varying degrees of curvature and height of folded peak in two opposing orientations (S1 Text and Fig 2A). To reflect real-world *in vivo* fluorescence imaging conditions, these synthetic data were corrupted with three incremental levels of Gaussian and Poisson noise (S1 Text and Fig 2A). Using the synthetic data, two types of error in computed polarity direction vectors were assessed: (1) an orientation flipping error, measured as the percentage of polarity vectors with an opposing orientation (opposite sign) to the polarity ground truth (S1 Text), and (2) direction accuracy, measured as the mean deviation angle between the polarity vectors produced by Origami, correctly oriented, and the polarity ground truth.

Of the two aspects of surface geometry analysed, height of folded peak (in two opposing directions) did not contribute significantly to orientation flipping errors (Linear Regression; $p = 0.86$, $R^2$ = -0.04). However, a larger radius of curvature of epithelium (a flatter epithelial sheet) did correlate with orientation flipping errors—albeit with a small effect of 0.08% increase for every 1 μm (5 pixels) increase in radius of curvature (Linear Regression; $p = 0.042$, $R^2 = 0.12$, effect), and a lower quality of segmentation output from ACME (Linear Regression $p < 0.001$, $R^2 = 0.46$; Fig 2B) computed as a Dice score. This meant a 0.2% reduction in Dice score for every 1 μm (5 pixels) increase in the radius of curvature. This correlation may be attributed to the reduced ability of ACME to segment flat, squamous cells in an epithelium oriented mostly along the lateral (*xy*) plane in data with anisotropic voxel resolution (here modelled using an anisotropic point spread function (PSF)). We found a correlation between noise applied to the synthetic images and errors in both polarity orientation flipping (ANOVA: $p \approx 0.001$; Tukey's contrasts showed 11.3% increase in errors at highest noise level compared with the lowest noise level applied: $p = 0.0039$) and segmentation output (ANOVA: $p < 0.01$; Tukey's contrasts showed 16.3% reduction in Dice score at highest noise level from the lowest noise level applied: $p = 0.0045$). Segmentation quality, in turn, influenced polarity orientation flipping, with errors below 1.5% at Dice scores above 0.8, but increasing with further decrease in Dice scores (Polynomial Regression; first-order: $p < 0.001$, Effect size = -28.78; second-order: $p < 0.01$, Effect size = 16.26; Fig 2C). Comparisons of many available segmentation algorithms when validating with fluorescent images from non-folded structures such as early-stage nematode embryos [16] or plant roots [18] have been shown to give Dice scores above 80%, suggesting a good performance under real experimental conditions.

Quantitative direction accuracy was evaluated in the synthetic data, for which, in contrast to data from real fluorescence images, a reliable ground truth could be generated from the known underlying surface functions. Compared to the polarity ground truth data, an overall offset of 10.6° ± 15.5° (mean ± std) was measured from our entire synthetic dataset. Just as for the polarity orientation flipping error, height of folded peak did not influence polarity direction accuracy (Linear Regression; $p = 0.39$, $R^2$ = -0.01), but there was a small effect of curvature of the epithelium with an additional 0.06° offset for every 1 μm (5 pixels) increase in radius of curvature of the epithelium (Linear Regression; $p = 0.005$, $R^2 = 0.24$). At the highest level of noise applied, errors in polarity orientation had a 6.6° greater offset than at the lowest noise level applied (Tukey's contrasts; $p = 0.003$). There was also a negative linear effect of

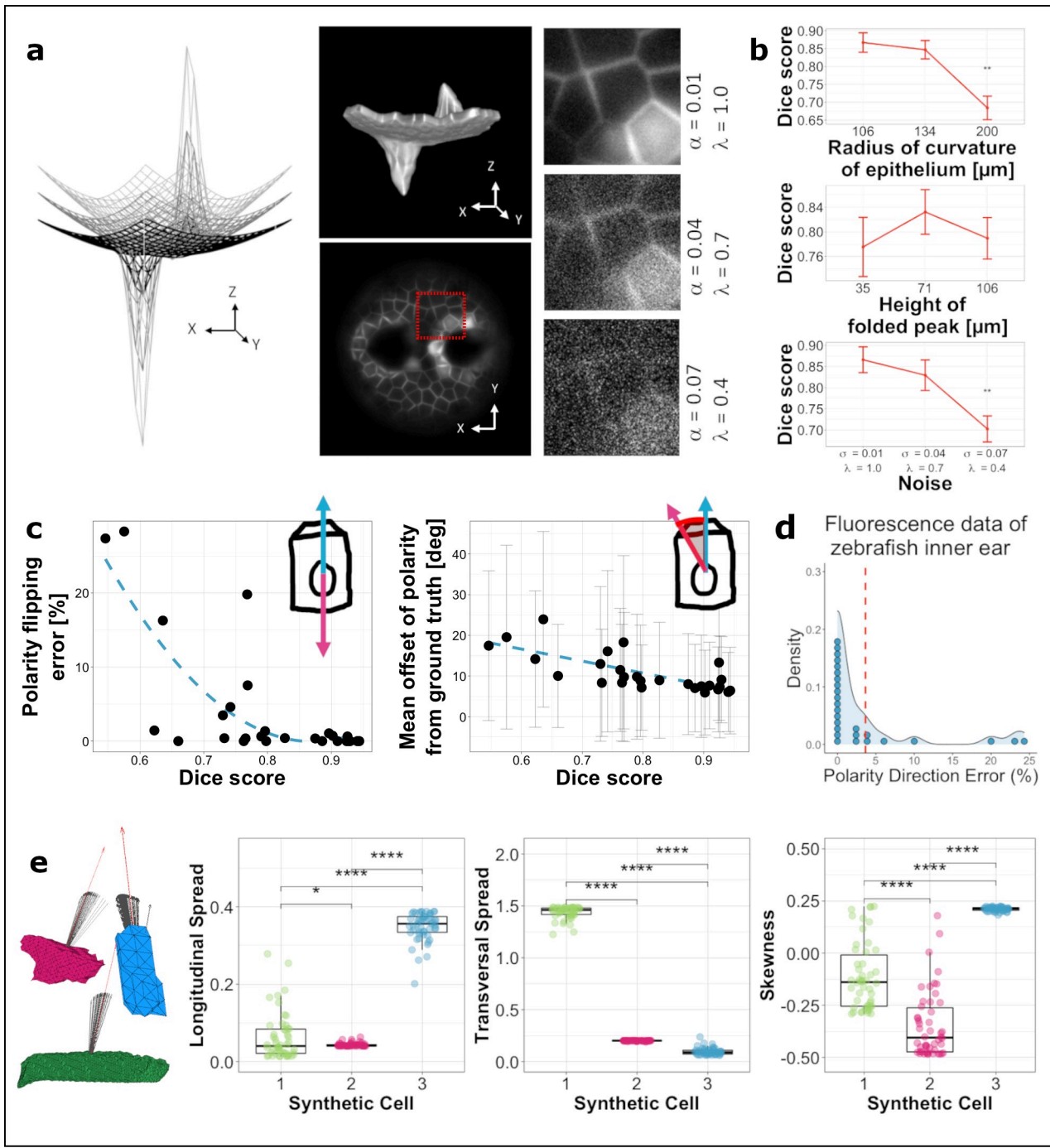

**Fig 2. Assessment of polarity assignment.** a. Surface meshes of synthetic epithelia for validating the Origami analysis pipeline. Alongside, 3D rendering of one of the synthetic epithelia (top) and a single 2D slice through it (bottom). Each image volume was corrupted with three levels of noise. b. The relationship between surface geometry/noise and segmentation quality. Error bars represent the standard deviation. Tukey's pairwise comparisons with significant values depicted with asterisks: Dice score at radius of curvature of 200 μm (1000 pixels) compared to that at 106 μm (530 pixels)–$p = 0.0004$, Dice score at largest noise level compared to the lowest: $p = 0.0045$. c. Effect of segmentation quality on errors in orientation flipping (left) and direction offset in the computed polarity vectors (error bars in grey represent standard deviation). Dashed lines represent quadratic and linear fit to data respectively. d. Probability density of errors in polarity direction in real fluorescence data from zebrafish embryos. Each dot represents the percentage error from a 3D segmented volume (n = 27; total of 949 segmented cells across all the images). The dashed line shows the mean error in the dataset (<4%). e. Sensitivity of cell shape metrics to errors in polarity orientation. Data points in the graphs are depicted with the same colour as the corresponding synthetic cell alongside. Tukey's pairwise comparisons with significant values depicted with asterisks; Longitudinal spread: 1–2 $p = 0.039$, 1–3 $p < 0.0001$, 2–3 $p < 0.0001$; Transversal Spread: 1–2 $p < 0.0001$, 1–3 $p < 0.0001$, 2–3 $p < 0.0001$; Skewness: 1–2 $p < 0.0001$, 1–3 $p < 0.0001$, 2–3 $p < 0.0001$.

segmentation quality with a 2.9˚ offset predicted for every 10% reduction in Dice score (Linear Regression; $p < 0.0001$, $R^2 = 0.53$; Fig 2C).

We further tested the effect of such errors in direction accuracy on the direction-variant shape metrics computed by applying directional noise—with a mean equal to the measured mean error above—to polarity vectors of three example cells showing extreme shape features from the synthetic dataset and computed direction-variant shape metrics for each new displaced polarity vector ($n = 50$; Fig 2E). The resulting computed shape metrics could still successfully differentiate between the three cells, showing that direction accuracy errors (excluding orientation flipping errors) should not adversely affect the shape metrics computed. On the other hand, orientation flipping errors will affect shape metrics, but as shown above, these errors are predicted to be small for a well-segmented image volume and can be easily identified by visual inspection and corrected if needed using the Origami pipeline.

Additionally, orientation flipping errors were quantified from real light-sheet fluorescence microscopy data from structures in the developing zebrafish otic vesicle (Figs 1 and 3). For this, cells assigned the wrong orientation along the apico-basal axis—that is, facing the basal surface instead of the apical surface, were identified by visual assessment in the Origami pipeline, showing errors in 3.65% of $n = 949$ cells analysed (Fig 2D).

## Proof of principle: Insights into cell shape dynamics during epithelial morphogenesis within the zebrafish inner ear

To further validate our method, we used Origami to characterise cell shape dynamics involved in the formation of different structures in the otic vesicle of the zebrafish embryo (Figs 1 and 3). We analysed light-sheet fluorescence image data from the anterior epithelial projection (AP) for the developing semicircular canal system, together with the endolymphatic sac (ES), at three developmental time points: 42.5 hours post fertilisation (hpf) (time point 1), 44.5 hpf (time point 2) and 50.5 hpf (time point 3), using three different fish for each time point. We also analysed the posterior epithelial projection (PP), a similar structure to the AP, but which develops later [40], at developmentally equivalent time points to that of the AP (46.5 hpf, 50.5 hpf and 60.5 hpf). The AP and PP are finger-like projections of the epithelium that move into the lumen of the vesicle, with the apical side of the cell on the outside of the curved projection surface [40]. By contrast, the ES forms as an invagination from dorsal otic epithelium, with the constricted apical surfaces of the cells lining the narrow lumen of the resultant short duct [8,45,46]. As the ES is formed through deformation of the epithelial sheet with opposite polarity to that of the epithelial projections, we expect cells in the ES and the projections to show significant differences in cell shape. Conversely, we do not expect significant differences in cell shape between the AP and PP cells, which form equivalent structures in the developing ear.

For each structure, the following shape attributes were computed at the single-cell level: surface area, sphericity, longitudinal spread, transversal spread and skewness. Since volume and surface area show high collinearity within our data (Pearson correlation coefficient = 0.98, 95% confidence intervals = [0.977, 0.984]), cell volume was excluded from further analysis. Although images included cells in the non-folding epithelium around the developing structures of interest, only cells from the folding epithelium were analysed. A multivariate analysis ({MANOVA.RM} package [47] in R [v 4.0.0]) of the dependence of cell shape attributes on the epithelial structure from which they are derived at different time points showed a significant difference between the three structures at the first two developmental times (Wald-type statistic; $p = 0.035$ (resampled $p = 0.001$) at time point 1 and $p < 0.001$ (resampled $p < 0.001$) at time point 2) but not at the final time point analysed ($p = 0.706$ (resampled $p = 0.038$)) for all shape attributes. Post-hoc Tukey's contrasts indicated that cells in the endolymphatic sac

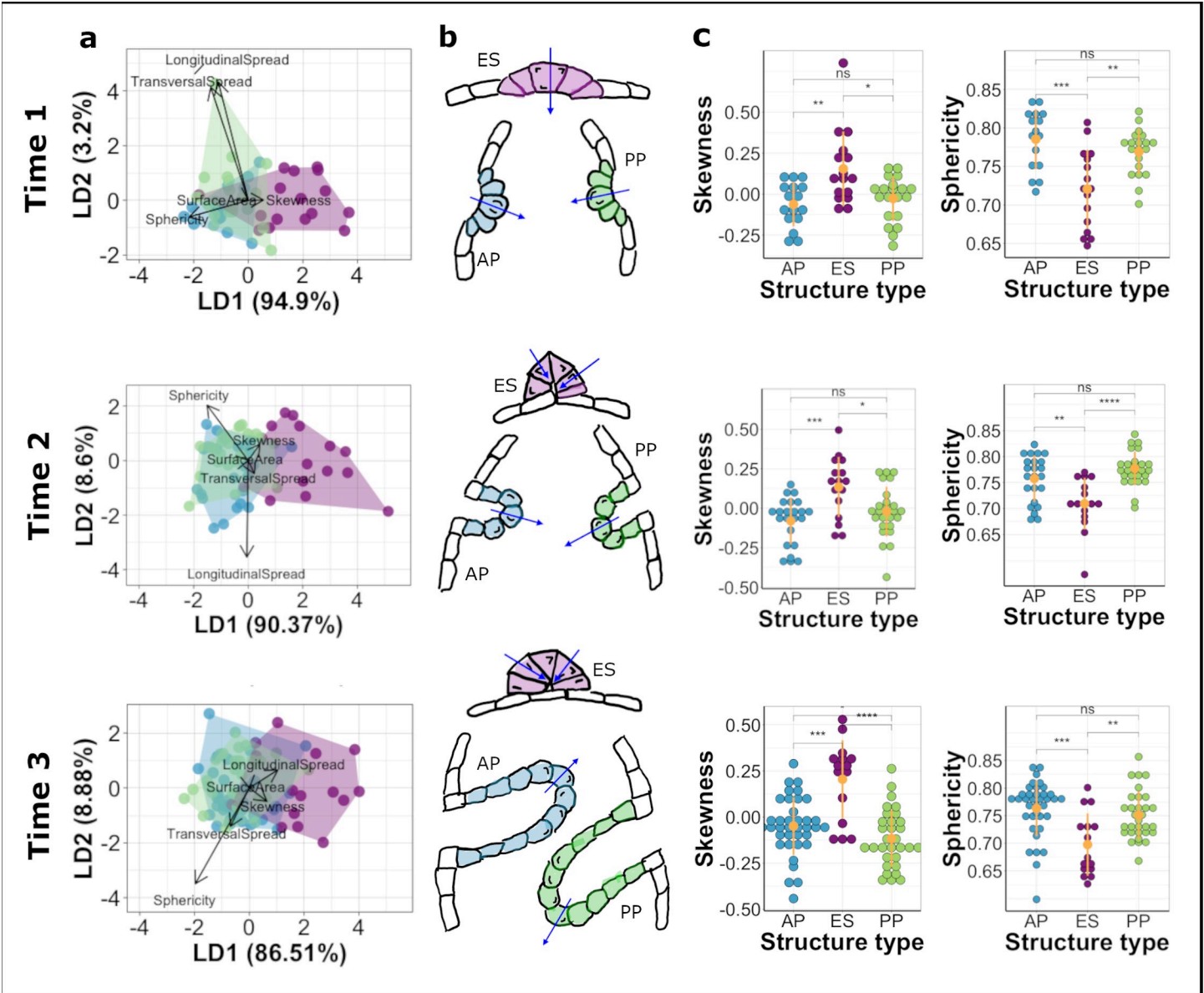

**Fig 3. Comparison of shape dynamics in developing structures of the zebrafish inner ear.** Rows represent each time point analysed. Data in blue represent cells from AP, green represent cells from PP and magenta represent ES. a. Linear discriminate analysis (LDA) biplots illustrate multivariate clustering of data—data from AP and PP show considerable overlap indicating similar shape signatures while data from ES show less overlap with the former. b. Schematic illustrations of cell shape signatures at the time points analysed showing cells in the ES having skew in the opposite direction to those in the projections and having less rounded shapes. Arrows indicate apico-basal polarity. c. Plots showing differences in skewness and sphericity between the structures at the time points analysed. Yellow dots with error lines represent mean and standard deviation for the data. *p* values for paired comparisons depicted are from Table 1.

showed significantly different shape dynamics from those of cells in both projections (ES—AP *p* = 0.006, PP—ES *p* = 0.012 at time point 1; ES—AP *p* = 0.0002, PP—ES *p* = 0.0002 at time point 2 but ES—AP *p* = 0.192, PP—ES *p* = 0.116 at time point 3). There was no significant difference in the cell shape signature between cells in the anterior and posterior projections (Tukey's contrasts; PP—AP *p* = 0.997 at time point 1; PP—AP *p* = 0.999 at time point 2 and PP—AP *p* = 0.896 at time point 3). These results indicate that the cell shape features included were more similar than different for cells from the structures at the third time point analysed.

**Table 1. Paired comparisons using Wilcoxon rank sum exact test (*p* values–adjusted using 'Holm' correction).**

|  | AP—ES | | | PP—ES | | | AP—PP | | |
|---|---|---|---|---|---|---|---|---|---|
| **Time point** | **1** | **2** | **3** | **1** | **2** | **3** | **1** | **2** | **3** |
| Skewness | **0.009** | **0.002** | **<0.001** | **0.036** | **0.036** | **<0.0001** | 0.42 | 0.286 | **0.047***  |
| Sphericity | **0.002** | **0.012** | **<0.001** | **0.005** | **<0.0001** | **0.007** | 0.149 | 0.228 | 0.07 |
| Surface Area | **0.001** | **<0.0001** | **0.032** | **0.001** | **<0.0001** | **0.013** | 0.887 | 0.85 | 0.912 |
| Transversal Spread | 0.057 | 0.062 | 1 | 0.357 | 0.062 | 1 | 0.357 | 0.897 | 1 |
| Longitudinal Spread | 1 | 0.88 | 0.054 | 1 | 0.81 | 0.102 | 1 | 0.46 | 0.582 |

*The differences in skewness between cells in the AP and PP at the 3rd time point tended towards significance. This might be attributed to differences in the lengths of projections, with cells at the leading end of the projection showing more extreme skewness values while cells along the lateral sides showing less skewed shape.

Of the attributes analysed, skewness (Kruskal-Wallis test; $p = 0.008$ at time 1, $p = 0.004$ at time 2 and $p < 0.0001$ at time 3), sphericity (Kruskal-Wallis test; $p < 0.001$ at time 1, $p = 0.00012$ at time 2 and $p < 0.001$ at time 3) and surface area (Kruskal-Wallis test; $p < 0.001$ at time 1, $p < 0.0001$ at time 2 and $p = 0.018$ at time 3) described significant differences in cell shape across all the three time points analysed; cells in the endolymphatic sac were characterised by positive skewness values, smaller sphericity values and larger surface areas as compared with cells in both projections, which show negative values of skewness (Table 1 and Fig 3).

The differences in surface area are likely to be attributed to differences in sphericity between the cells in the three structures, but not in dimensions, as the transversal and longitudinal spread showed no significant differences.

## Availability and future directions

Origami is free to download from: https://github.com/cistib/origami. It is implemented within MATLAB (compatibility with version 2018b onwards) and includes additional tools for visualising cell shape metrics from complex folding epithelia at the single-cell level. Instructions for installation and use are included with the software.

Our software can accept pre-segmented data, making it compatible with segmentation algorithms of the user's choice, potentially allowing for data acquired using other 3D imaging techniques such as tomography to be analysed. Segmented data must represent cell shape accurately, and so the choice of imaging technique that can faithfully detect 3D cell shape alongside membrane or cytoplasm-based segmentation is critical.

We used *a priori* knowledge of the otic epithelium organisation to inform the orientation of the apico-basal axis of the epithelial sheet to face the lumen of the otic vesicle [2,8,40]. It is essential to know the organisation of the apico-basal axis of cells within any new structure studied to apply Origami—wherein, the orientation-determining parameter can then be set to always be 'in' or 'out' depending on if the polarity direction vector is required to point towards the inside or outside face of a curved structure. We also assumed that individual cells do not violate this organisation, as this cannot be detected without additional polarity-specific labels. In such a case, polarity vectors from our analysis can be complemented with information from polarity-specific labelling to track such behaviour. Moreover, to compute shape features along an alternative axis of polarity, the pipeline can accept pre-assigned polarity as a cell-specific vector-list.

We expect Origami to be applied to studying a wide range of morphogenetic processes and to contribute to our understanding of the biomechanical processes underpinning them.

## Supporting information

**S1 Software Code. Zip file containing MATLAB scripts and instructions for installing and running Origami software.** Requires MATLAB (v 2018b onwards).
(7Z)

**S1 Text. Text file detailing methodology used for collection of fluorescence microscopy data, generation of synthetic membranes and parameters used for membrane segmentation.**
(DOCX)

## Acknowledgments

We thank N. van Hateren for assistance with imaging, S. Burbridge and M. Marzo for technical support, and the Sheffield Aquarium Team for zebrafish husbandry.

## Author Contributions

**Conceptualization:** Sarah Baxendale, Tanya T. Whitfield, Alejandro F. Frangi.

**Data curation:** Tania Mendonca, Ana A. Jones, Sarah Baxendale, Alejandro F. Frangi.

**Formal analysis:** Tania Mendonca.

**Funding acquisition:** Sarah Baxendale, Tanya T. Whitfield, Alejandro F. Frangi.

**Investigation:** Sarah Baxendale, Tanya T. Whitfield, Alejandro F. Frangi.

**Methodology:** Tania Mendonca, Ana A. Jones, Jose M. Pozo, Sarah Baxendale, Tanya T. Whitfield, Alejandro F. Frangi.

**Project administration:** Tania Mendonca, Sarah Baxendale, Tanya T. Whitfield, Alejandro F. Frangi.

**Resources:** Sarah Baxendale, Tanya T. Whitfield, Alejandro F. Frangi.

**Software:** Tania Mendonca, Jose M. Pozo.

**Supervision:** Tanya T. Whitfield, Alejandro F. Frangi.

**Validation:** Tania Mendonca, Jose M. Pozo.

**Visualization:** Tania Mendonca.

**Writing – original draft:** Tania Mendonca.

**Writing – review & editing:** Tania Mendonca, Ana A. Jones, Jose M. Pozo, Sarah Baxendale, Tanya T. Whitfield, Alejandro F. Frangi.

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
