## [Decision Letter · Decision Letter 0]

17 Jul 2021

Dear Dr Mendonca,

Thank you very much for submitting your manuscript "Origami: Single-cell oriented 3D shape dynamics of folding epithelia from fluorescence microscopy images" for consideration at PLOS Computational Biology.

As with all papers reviewed by the journal, your manuscript was reviewed by members of the editorial board and by several independent reviewers. In light of the reviews (below this email), we would like to invite the resubmission of a significantly-revised version that takes into account the reviewers' comments.

Specifically,please address the reviewers' concerns regarding the applicability of Origami, benchmarking vs. other methods, and the availability of source code repository. Publication in the software section requires that the code is deposited in the code repository, such as github along with examples and user tutorial.

We cannot make any decision about publication until we have seen the revised manuscript and your response to the reviewers' comments. Your revised manuscript is also likely to be sent to reviewers for further evaluation.

Sincerely,

Dina Schneidman

Software Editor

PLOS Computational Biology

Specifically,please address the reviewers' concerns regarding the applicability of Origami, benchmarking vs. other methods, and the availability of source code repository. Publication in the software section requires that the code is deposited in the code repository, such as github along with examples and user tutorial.

Reviewer's Responses to Questions

**Comments to the Authors:**

Reviewer #1: Review of Mendonca et al., “Origami: Single-cell oriented 3D shape dynamics of folding epithelia from fluorescence microscopy images”

In this manuscript, the authors describe new MATLAB-based software to characterize and quantify 3D epithelial cell shape, with only prior knowledge of epithelial orientation. Because many prior approaches require some knowledge of relevant shape features, this problem has not been generally solved. Origami provides a useful step forward in the efforts to capture and quantify meaningful individual cell features within an organized tissue. My comments are mainly minor general questions related to the application of the software to this and other tissues.

• It would be helpful if the authors could comment on different imaging modalities and optimization of imaging parameters. Although segmentation can be carried out prior to analysis using Origami, segmentation seems to be a potential weak link during the Origami validation using synthetic data. Did the authors optimize the imaging for ACME? When using laser scanning confocal microscopy, was the Airyscan modality used? Was it necessary?

• For the timecourse data, I am not sure if I missed this, but how many embryos were used for each timepoint?

• Did the authors test how consistent the data are between wild type embryos at any one individual timepoint? This might help to establish the kind of noise one might expected in the data, as cells and embryos are not exactly alike.

Reviewer #2: The manuscript presents Origami, a microscopy image analysis pipeline devoted to the quantification of shape in oriented biological structures. Origami relies on a mesh representation of segmented objects, and then computes shape features from the 3D geometrical moments introduced by a subset of the authors in a 2011 IEEE PAMI paper.

I have major concerns about three aspects of this work:

1) Novelty: the technical/methodological/algorithmic novelty of Origami is not significant and in my opinion too minor to fit the scope of PLOS CompBio. Providing a freely-available MATLAB implementation of a compilation of a bunch of existing methods (triangulation and meshing, mesh cleaning/processing, geometrical moments) is definitely useful but of limited computational innovation. Although there doesn't seem to be any, if novel computational methods had to be developed for this pipeline (as opposed to assembling existing ones), the authors should highlight them more clearly.

2) Significance: although Origami is described as a way to quantify cell shape features in general, it entirely relies on prior knowledge of the epithelium organisation. As such, it can only provide oriented shape features when orientation is known a priori, which makes it limited to very specific settings. The introduction is particularly confusing as it emphasizes the issue of orienting cells along biologically relevant axis, which *is precisely not* solved by Origami. Origami seems to focus mostly on implementing a new method to extracts shape features along a known polarity axis, which can per se be valuable but it not spelled out clearly here. The authors should work on a major rewrite to a) define the scope of their work better and b) outline which problem Origami is aiming to solv, and c) clarify what Origami can bring over other approaches (see 3)).

3) Comparison/benchmarking: the authors dismiss several related methods on the basis that they are "specific to the geometry of the specimen"/"applicable only if a shape feature is known". I do not think this argument is valid since the whole orientation/polarity estimation of Origami relies precisely on the known geometry/features of the considered epithelium. Comparative experiments must be carried out to quantify how much the shape features extracted by Origami bring over those extracted eg by LongAxis. A sensible experiment could for instance be to orient the "long axis" of the cells along the known apico-basal axis, and compare the robustness of the features extracted by LongAxis and Origami. An additional important aspect to investigate is the relevance of the mesh representation. How would the skewness, transversal/longitudinal spread, sphericity and surface area features of a simple ellipse fitted on the voxel data of the segmented cell such that its major axis is aligned to the known apico-basal axis fare against the geometrical moments proposed here?

On a more minor note, I would encourage the authors to make their code available on a github repository as this would facilitate traceability and reuse.

Reviewer #3: This manuscripts present a MATLAB program to find the apical-basal axis of each cell in an arbitrarily curved epithelia, and then calculates cell-based shape metrics based on this axis. Shape information of a cell with respect to its apical-basal axis is an essential first step in trying to understand and model many aspects of epithelial morphogenesis. There is some existing work in the literature but not a generally useful approach so the current approach fills an important gap. The chosen method is straightforward and good (fits a surface mesh to the centroids of all cells and then calculates surface normals for all faces after handling mesh defects). The manuscript is generally clearly written and explains the approach at the right level of detail. The method is nicely validated using appropriate metrics on synthetic data and real data from the zebrafish inner ear.

I had a few minor concerns:

Abstract/title should be more specific as to what “orientation” means- apical-basal, planar cell polarity, major axis of a cell are all possible interpretations of “orientation”. Should include “apical-basal” in abstract and maybe title to clearly indicate what the point of the paper is.

Calculation of moments. It appears the authors are using the vertex positions of the surface mesh to calculate a cell’s moments. This could run into errors if the mesh is not regular (e.g. smaller faces/more vertices in one part than another. There is also the issue of calculating moments of the surface shell rather the whole volume which would give unexpected results on cells with thin processes such as neurons or pseudostratified epithelia. I think it would be better to convert the mesh into an image—i.e. fill it with isometric voxels and then compute the moments using the set of voxels.

**Have the authors made all data and (if applicable) computational code underlying the findings in their manuscript fully available?**

Reviewer #1: Yes

Reviewer #2: Yes

Reviewer #3: Yes

PLOS authors have the option to publish the peer review history of their article (what does this mean?). If published, this will include your full peer review and any attached files.

Reviewer #1: No

Reviewer #2: No

Reviewer #3: No
---

## [Decision Letter · Decision Letter 1]

13 Oct 2021

Dear Professor Frangi,

We are pleased to inform you that your manuscript 'Origami: Single-cell 3D shape dynamics oriented along the apico-basal axis of folding epithelia from fluorescence microscopy' has been provisionally accepted for publication in PLOS Computational Biology.

Additionally, please address the comments of the 2nd reviewer in the final version.

Best regards,

Dina Schneidman

Software Editor

PLOS Computational Biology

Reviewer's Responses to Questions

**Comments to the Authors:**

Reviewer #1: In this revision, the authors have responded well to all of my questions and concerns. As I stated in my initial review, the problem that the authors are addressing, how to characterize and quantify epithelial cell shape in 3 dimensions, has not been generally solved. Origami may be useful for many investigators working a variety of tissues where cell shape cannot be easily generalized.

Reviewer #2: I thank the authors for their response to my comments and their work in revising the paper.

I do entirely agree that consolidating tools into easy-to-use pipelines is immensely valuable, as acknowledged in my first review. I was instead challenging whether the consolidation effort presented in this work fits within PLOS CompBio's scope, and consists of "enhancements to an existing published methods that bring exceptional new capabilities" (https://journals.plos.org/ploscompbiol/s/journal-information). As much as I agree on the principle that requiring algorithmic novelty pushes for over-engineering, PLOS CompBio lists originality and innovation as part of its criteria for publication, and I therefore do not think that asking for the algorithmic novelty of the work to be better outlined and stated is unreasonable. It would be helpful to state right away in the introduction that the main novelty of Origami lies in is its easy-to-use implementation and not in its core algorithm, as it is currently still not made explicit.

The clarification of direction VS orientation and the associated modifications in the manuscript are very helpful and clarify several aspects I had misunderstood. I think it would be useful to have a few sentences at the end of the paper discussing how the identification of the sign of the direction vector (what is now called the orientation), that is not automatically extracted by Origami, could be automated or corrected. In their response, the authors mention that it can be manually edited - it would be useful to discuss how this could be done concretely and whether Origami provides any sort of support to help with that.

Finally, I immensely appreciate that the authors now provide their code on github, as this will facilitate reuse and dissemination.

**Have the authors made all data and (if applicable) computational code underlying the findings in their manuscript fully available?**

Reviewer #1: Yes

Reviewer #2: Yes

PLOS authors have the option to publish the peer review history of their article (what does this mean?). If published, this will include your full peer review and any attached files.

Reviewer #1: No

Reviewer #2: No

---

## [Editor Report · Acceptance letter]

28 Oct 2021

PCOMPBIOL-D-21-00819R1 

Origami: Single-cell 3D shape dynamics oriented along the apico-basal axis of folding epithelia from fluorescence microscopy data

Dear Dr Frangi,

I am pleased to inform you that your manuscript has been formally accepted for publication in PLOS Computational Biology. Your manuscript is now with our production department and you will be notified of the publication date in due course.

With kind regards,

Andrea Szabo
